# Genetics and Epigenetics of Spontaneous Intracerebral Hemorrhage

**DOI:** 10.3390/ijms23126479

**Published:** 2022-06-09

**Authors:** Eva Giralt-Steinhauer, Joan Jiménez-Balado, Isabel Fernández-Pérez, Lucía Rey Álvarez, Ana Rodríguez-Campello, Ángel Ois, Elisa Cuadrado-Godia, Jordi Jiménez-Conde, Jaume Roquer

**Affiliations:** Neurovascular Research Group, Neurology Department, Hospital de Mar, 08003 Barcelona, Spain; ifernandezperez@psmar.cat (I.F.-P.); lrey@imim.es (L.R.Á.); arodriguezc@psmar.cat (A.R.-C.); aois@psmar.cat (Á.O.); ecuadrardo@psmar.cat (E.C.-G.); jjimenez@imim.es (J.J.-C.); jroquer@psmar.cat (J.R.)

**Keywords:** intracerebral hemorrhage, genetics, epigenetics

## Abstract

Intracerebral hemorrhage (ICH) is a complex and heterogeneous disease, and there is no effective treatment. Spontaneous ICH represents the final manifestation of different types of cerebral small vessel disease, usually categorized as: lobar (mostly related to cerebral amyloid angiopathy) and nonlobar (hypertension-related vasculopathy) ICH. Accurate phenotyping aims to reflect these biological differences in the underlying mechanisms and has been demonstrated to be crucial to the success of genetic studies in this field. This review summarizes how current knowledge on genetics and epigenetics of this devastating stroke subtype are contributing to improve the understanding of ICH pathophysiology and their potential role in developing therapeutic strategies.

## 1. Genetics and Epigenetics of Spontaneous Intracerebral Hemorrhage

Spontaneous intracerebral hemorrhage (ICH) accounts for approximately 15% of stroke cases and is still a considerable source of neurological morbidity and mortality. Given the increase in life expectancy and widespread use of antithrombotic therapy in the elderly, the incidence of ICH is expected to rise in coming years [1,2]. Primary ICH refers to the rupture of damaged arteries or arterioles and is the final manifestation of different types of cerebral small vessel disease that progress sub clinically over several years before the brain bleed occurs [3]. Although primary ICH might be responsible for around 80% of cases of non-traumatic ICH, clinicians should consider searching for other causes of ICH (coagulopathy, vascular malformation rupture, cavernous malformation, moyamoya, vasculitis, tumor, cerebral venous thrombosis, among others), also called secondary ICH, beyond the scope of the present review.

Histopathological observations (corroborated by epidemiologic, neuroimaging, and genetic studies) proved that the main underlying vessel disease differs according to the location of the bleeding in the brain, and consequently, primary ICH can be classified into two main categories: nonlobar and lobar [4].

Nonlobar ICH originates in deep brain structures (basal ganglia, thalamus, brainstem, and deep cerebellum) and has been consistently associated with hypertensive induced vasculopathy [5,6]. On the other hand, lobar ICH (located at the cortical region or at the junction between cortex and white matter) is mostly related to cerebral amyloid angiopathy (CAA), in which β-amyloid accumulates within leptomeningeal and intracortical blood vessels, causing a reduction in the number of smooth muscle cells and fibrinoid necrosis, which ultimately results in vessel damage, rupture, and bleeding [7,8]. Distinguishing between CAA-related lobar ICH and hypertensive lobar ICH is complex but bears prognostic relevance due to the risk of recurrence and dementia, which are both significantly higher in the CAA-related lobar ICH [9,10,11].

Age is one of the main risk factors of ICH, such that the relative risk of developing ICH doubles for every 10-year increase in age. Another risk factor is sex at birth, being more common in men than in women. Additionally, the incidence of ICH varies greatly depending on the ethnicity. For instance, the ICH incidence in Asian populations doubled that seen in Caucasians [12]. Hypertension is the main modifiable risk factor for both nonlobar and lobar ICH [13,14], although it shows a greater effect for nonlobar ICH [15]. The incidence of ICH increased progressively with increasing blood pressure [16]. Additional risk factors include current cigarette smoking and heavy alcohol consumption, which may induce coagulopathy and platelet dysfunction and directly affects cerebral vessel integrity [17]. Reduced concentrations of total cholesterol, low non-high-density lipoprotein cholesterol, and increased high-density lipoprotein cholesterol seems to be associated with an increased risk of ICH [13,18].

The pathophysiological mechanisms after ICH are complex and not fully understood, but a two-phase model of brain damage has been proposed [19,20]. The primary injury occurs within the first hour after ICH onset, when the blood released into the brain parenchyma leads to mechanical compression of anatomical structures and increased intracranial pressure, even herniation and death. The secondary brain injury is subsequently triggered by the activation of resident cells, the infiltration of peripheral immune cells, and the secretion of inflammatory factors that induce a cascade reaction, which will eventually lead to pathological changes such as BBB disruption, oedema, and cell death [21]. Multidisciplinary ICH management (including rapid identification, medical treatment, and acute neurosurgical intervention—when indicated) is essential to facilitate recovery, but an effective strategy for prevention or treatment of secondary brain injury is still undetermined [10,22,23].

Long-term functional independence is achieved in only 12–39% of cases, with mortality rates of 54% at 1 year [12]. Patients who survive hemorrhagic stroke have a continuing elevated risk of death compared with matched individuals from the general population [24]. Therefore, the identification of biological pathways that could potentially be targeted as new therapies are urgently needed to reduce the health care burden associated with this devastating neurologic disease.

## 2. ICH Genomics

Although the risk factors mentioned above explain partially the variance in ICH risk, a significant portion of this variation remains unexplained. Several monogenic disorders that follow a Mendelian pattern of inheritance manifest clinically through ICH. These forms account for less than 1% of all ICH cases and tend to appear at younger ages and preferentially affect Caucasians [22]. Two examples of Mendelian ICH include familial cerebral amyloid angiopathy [8] and COL4A1-related ICH [25].

The familial aggregation of ICH has been documented in several studies [26], and individuals with a first-degree relative with ICH showed a six-fold increased risk for both lobar and nonlobar ICH [27]. In 2013, pseudo-heritability for overall ICH risk, derived from a genome-wide association study (GWAS), was firstly estimated to be 44%. This study found that 73% and 34% of variance in the risk of lobar and deep ICH, respectively, is explained by genetic risk factors [28].

However, most frequent genetic findings in ICH involve a non-mendelian contribution. The genetics of ICH was traditionally approached by candidate-gene association studies, which test the hypothesis that common genetic variants are associated with a disease of interest, usually focusing on potential environmental risk factors (e.g., mutations in genes related to blood pressure and lipid levels) or possible underlying pathogenic pathways (for example vascular wall integrity, endothelial function, vessel wall reactivity, or coagulation) [29]. Many of these candidate gene studies, limited by the small sample size and biased in different ways, have not been robustly replicated, leading to a paradigm shift.

Advances in genotyping technology have led to the possibility to agnostically screen millions of genetic markers across the genome [30]. Genome-wide association studies aim to identify common genetic variants associated with the outcome of interest by using preselected single nucleotide polymorphisms (SNPs) genotyping chips that can detect 0.5–2.5 million of the 38 million known genetic variants [31]. Thanks to linkage disequilibrium (LD) and imputation, the number of evaluated variants increases to 5–15 million.

Data can include complete reads of DNA, either for coding regions (whole-exome sequencing, 1% of the genome) or the entire genome (whole-genome sequencing). The major advantage of these sequencing strategies is that they may identify rare variants, including de novo mutations that are undetectable by SNP arrays. This approach yields substantially more data than SNP arrays but is more expensive and much of these data might be non-informative [32].

Huge efforts have been made in the last decade in to improve the patients’ phenotyping and achieving appropriate sample size, given ICH heterogeneous biology and low incidence. The creation of large international consortia has led to substantial progress in this field.

### 2.1. Genome-Wide Association Studies (GWAS) in ICH

ICH is a complex disease, and the underlying genetic model is believed to be multifactorial: on one hand, numerous genetic polymorphisms would confer small individual increases in ICH risk. On the other hand, environmental risk factors, also having relatively small effects, would confer additional risk of ICH, individually or in interaction with genetic risk factors [33,34].

Therefore, in this review, we provide a summary of the most significant findings to date in the field of GWAS, which reported several SNPs that are robustly associated with ICH (Figure 1).

### 2.2. Examples of Genetic Risk Factors for ICH

#### 2.2.1. APOE Alleles

*APOE* is a polymorphic gene located on chromosome 19 with two common SNPs: 388 T > C (rs429358) and 526C > T (rs7412). These two SNPs produce the following alleles: *Ɛ**2*, *Ɛ**3* and *Ɛ**4*, having allele frequencies of approximately 7%, 81%, and 14%, respectively [42]. The individual *APOE* genotype emerges from the combination of two alleles (for example, *Ɛ**2/**Ɛ**4*), which encode one of the three major isoforms of the apolipoprotein E (ApoE Ɛ2, ApoE Ɛ3, and ApoE Ɛ4), a polymorphic glycoprotein involved in lipid transport and in cell membrane maintenance [43,44]. 

Both *APOE*
*Ɛ**2* and the *APOE*
*Ɛ**4* alleles are independent risk factors for lobar ICH at a “genome-wide significance level”, with odds ratios ranging between 1.8 (*p* = 6.6 × 10^−10^) and 2.2 (*p* = 2.4 × 10^−11^), respectively, using *APOE*
*Ɛ**3* homozygosity as the reference category [39]. The *Ɛ**4* allele increases the severity of vascular amyloid deposition, whereas *Ɛ**2* is associated with pathological signs of increased vessel damage due to amyloid deposition, such as concentric vessel splitting and fibrinoid necrosis, which lead to vessel rupture [45]. For lobar ICH, a genetic association study showed that *APOE*
*Ɛ**2* carriers had larger ICH volumes. Specifically, each allele copy increased the hematoma size by 5.3 cc (95% confidence interval (CI) 4.1–6.2 cc, *p* = 0.004), which has been replicated in Europeans and African Americans, resulting in a 50% increased risk of poor outcome [46]. A suggested mechanism explaining these results is that the blood vessels in the brain surrounding the initial bleed are more likely to bleed themselves (since they have a more severe pre-existing pathology) and contribute to propagation of the hematoma [47].

In a recent meta-analysis, *APOE*
*Ɛ**2* and *Ɛ**4* alleles remain genetic risk factors for lobar ICH, but these results are largely driven by the associations in white individuals. *APOE*
*Ɛ**2* and *Ɛ**4* did not show an association with nonlobar ICH risk in any race/ethnicity [40].

#### 2.2.2. Nonlobar ICH Genetic Risk: 1q22

In 2014, the International Stroke Genomics Consortium completed the first GWAS of ICH, and identified 1q22 as the first nonfamilial genetic risk locus for nonlobar ICH, and it was replicated in an independent sample. The identified genetic risk variants are located in a region that contains the *PMF1* gene (codes for polyamine modulated factor 1, a protein needed for normal chromosome alignment and segregation and kinetochore formation during mitosis) and *SLC25A44* (codes for a mitochondrial carrier protein). The highest association corresponded to rs2984613 [4]. Moreover, this locus was also found to be associated at a genome-wide significant level with an increased white matter hyperintensity burden [48], highlighting the plausible pathophysiological link between both entities, as an expression of cerebral small vessel disease.

#### 2.2.3. Risk Prediction Using Genetic Information

Genetic risk scores (GRS) combine genomic data to sum up the effects of several SNPs associated with a determined phenotype to investigate the aggregate effect of the polymorphisms. The creation of a risk score helps to stratify the cumulative genetic risk of ICH at the individual level.

In 2012, Falcone and collaborators published a blood pressure (BP)-based GRS, which included thirty-nine SNPs reported to be associated with blood pressure levels. The BP-GRS was associated with the risk of ICH (OR 1.11; 95% CI 1.02–1.21; *p* = 0.01), although this result was driven by the subset of patients with deep ICH (OR 1.18; 95% CI 1.07–1.30, *p* = 0.001), while lobar ICH risk showed no association with the BP-based GRS [49].

In another study, using a 42 build BP-GRS, the aggregate burden of these risk alleles was associated with larger hematoma volume in nonlobar ICH, and poor functional outcome at 90 days in all (considering both together) and nonlobar ICH. Concretely, for each standard deviation increase in the score, the authors observed a 39% increase in the risk of having a poor clinical outcome. This effect, however, seemed to be driven by the embedded effect of the GRS on non-lobar ICH [50].

Similarly, Falcone and collaborators have recently created a lipid trait (total cholesterol, LDL, high-density lipoprotein, and triglycerides) GRS using independent genome-wide significant SNPs for each trait, such that the GRS for total cholesterol (OR 0.92; 95% CI 0.85–0.99; *p* = 0.03) and LDL (OR 0.88; 95% CI 0.81–0.95, *p* = 0.002) were both inversely associated with ICH risk [51].

Several intermediate phenotypes are related to ICH risk as well, and it is thus of interest to know if a SNP or combination of multiple SNPs for these phenotypes are also associated with an increased risk of ICH, either independently or mediated by the phenotype [52]. These analyses, called Mendelian randomization (MR), can provide evidence of causality. Moreover, the identification of these genetic variations that are shared between stroke and other complex phenotypes can also shed light on ICH mechanisms [53]. 

For instance, in a recent study, Chung and collaborators (2019) aimed to identify novel genetic risk associations for ICH (nonlobar ICH, lobar ICH, and all ICH) using cross-phenotype analyses with GWAS summary statistics of small vessel ischemic stroke (SVS). Thereby, the authors identified novel loci at 2q33 and 13q34 associated with nonlobar ICH. The genes at these loci have been implicated in other common diseases or traits related to cerebral small vessel disease (CSVD) and are now implicated in ICH as well, demonstrating pleiotropic associations with small vessel ischemic stroke because both are manifestations of hypertensive CSVD [48].

MR analyses performed in the above-mentioned study of lipid trait analysis and ICH indicated that a 1 mmol/L (38.67 mg/dL) increase in genetically instrumented total and LDL cholesterol was associated with 23% and 41% lower risks of ICH, respectively, providing support for a potential causal role of LDL cholesterol in ICH [51].

GRS, MR, and a combination of both, are some of the methods developed in order to translate major recent discoveries in ICH genetics into clinical practice.

## 3. ICH Epigenomics

Epigenetics includes modifications able to change gene expression without altering the DNA sequence, and it is usually considered the interface between genome and environment [54]. It is a promising field of growing interest, since epigenetic modifications are reversible processes, triggered by lifestyle and nutritional factors. In recent years, epigenetic changes have been related to the pathogenesis of cardiovascular diseases including ischemic stroke [55]. The epigenetic contribution to ICH is still largely unknown, but some evidence has emerged in recent years. Several preclinical studies appear promising utilizing agents that target epigenetic pathways but with unsuccessful results in clinical settings [56]. The main epigenetic modulations include DNA methylation, noncoding RNAs, and histone modifications.

### 3.1. DNA Methylation (DNAm)

DNAm, the most broadly studied epigenetic modification, inhibits gene expression through transcriptional silencing. DNAm is the reversible addition of a methyl group to the 5-carbon of cytosine in a cytosine-phosphate-guanine site (CpGs) to form a 5 methylcytosine, mediated by DNA methyltransferases. This dinucleotide is quite rare in mammalian genomes (1%) and clusters in regions known as CpG islands [57]. As Global DNA methylation (GDM) patterns changes over time, a loss of genomic DNA methylation has been found in a variety of common aging-related diseases [58]. Aberrant GDM has been associated with atherosclerosis, coronary heart disease, and hypertension [59,60,61]. It has also been reported that ischemic stroke patients show global DNA hypomethylation compared with healthy individuals [55,58]. However, less is known about the role of DNAm in ICH. In a recent study, GDM patterns in whole blood from ICH patients (*n* = 30) and controls (*n* = 34) were compared. The ICH and control groups showed significantly different DNAm patterns at 1530 sites (*p* < 5.92 × 10^−8^), with 1377 hypermethylated sites and 153 hypomethylated sites in ICH patients compared to the methylation status in healthy controls. Some of these differentially methylated sites were associated with promoters and genes related to inflammatory pathways [62].

### 3.2. RNA-Based Epigenetic Mechanisms

Recent studies have shown that non-protein-coding DNA sequences, which constitute more than 98% of the human genome, are pervasively transcribed, forming numerous subtypes of non-protein-coding RNAs (ncRNAs) [63]. Further investigations of these ncRNAs have demonstrated the existence of RNA-based networks that are involved in regulating nearly every cellular process [64]. These ncRNAs include subclasses that have been recently characterized, such as microRNAs (miRNAs), long non-coding RNAs (lncRNAs), and circular RNAs (circRNAs).

MiRNAs represent the best-characterized subclass of ncRNAs. They are endogenous small (typically consisting of 18–25 nucleotides) RNA molecules that regulate the expression of hundreds of target genes via sequence-specific interactions with messenger RNA (mRNA) [65]. Moreover, some miRNAs that are differentially expressed in brain tissue can be detected in peripheral blood as well, suggesting that these small ncRNA molecules may be involved in mediating systemic responses to cerebral damage [65,66]. Some clinical studies found differentially expressed miRNAs between ischemic and hemorrhagic stroke subtypes, although only miRNA-126 and miRNA-146a were replicated [67], despite substantial methodological heterogeneity.

In rat models, the overexpression of miRNA-126 showed a protective role in ICH by attenuating intracerebral hemorrhage-induced blood–brain barrier disruption, which is associated with down-regulated expression of VCAM-1 in the hemorrhagic area [68], and exhibits an anti-apoptotic effect [69]. Preclinical studies in different neurological disorders showed miRNA-146a to be an important repressor of inflammation [70] and to promote oligodendrogenesis [71]. Regarding patients with ICH, some miRNAs (Table 1) are differentially expressed with respect to controls.

In conclusion, further investigations are required to evaluate the potential of miRNA as biomarkers, to confirm their roles in more diverse populations and to elucidate their contributions by conducting more functional experiments.

On the other hand, the lncRNAs, RNA molecules comprising more than 200 nucleotides, have gained widespread attention in recent years. Although lncRNAs were primarily considered simply transcriptional by-products, recent evidence suggests that lncRNAs represent a new and crucial layer of biological regulation [77,78]. They seem to modulate gene expression at epigenetic, transcriptional, post-transcriptional, and chromatin remodeling levels by directly binding to the target genes or recruiting transcription factors [79]. In a study by Yang et al., key lncRNAs were identified in competing endogenous RNA networks, which were validated through real-time qPCR with peripheral blood samples from patients with ICH (*n* = 33) compared with controls subjects (*n* = 15). Six lncRNAs (LY86-AS1, DLX6-AS1, RRN3P2, and CRNDE were downregulated, while HCP5 and MIAT were upregulated) showed to be differentially expressed in patients with ICH compared to controls subjects [80].

CircRNAs, a heterogeneous group of noncoding transcripts with covalent bonds between head 3′ and tail 5′ ends to cause a circular pattern, are abundantly expressed in the central nervous system [79]. Bai et al. aim to investigate the expression profiles of circRNAs after ICH (*n* = 44). Expression levels of three circRNAs (has_circ_0001240 and has_circ_0001947 were upregulated, and has_circ_0001386 was downregulated) were consistently altered (validated in an independent sample) in patients with ICH compared with their expression levels in patients with hypertension. Functional analysis in this study showed circRNAs to be mainly involved in fatty acid biogenesis, lysine degradation, integrin cell surface interactions and the immune system [81].

Understanding complex epigenetic mechanisms, such as RNA-based networks, will provide many future strategies that exploit these processes for treatment of complex neurological disease such as ICH [65].

### 3.3. Histone Modifications

DNA associates with histone proteins in subunits called nucleosomes that form chromatin. Histones can undergo a number of covalent post-translational modifications, including acetylation and methylation. These modifications modulate histone–DNA interactions by influencing the chromatin structure, thereby directing the accessibility of transcriptional regulators to DNA-binding elements [82]. Histone acetylation catalyzed by histone acetyltransferases usually increases gene expression, whereas its deacetylation, performed by histone deacetylases (HDACs), inhibits gene expression. On the other hand, histone methylation can either activate or inhibit the expression of genes [83,84]. Despite the positive results obtained in preclinical models [85], the use of HDAC inhibitors have not yet provide satisfactory results in patients with ICH.

## 4. Exome Wide Association Studies (EWAS)

Exome sequencing is an efficient strategy to selectively sequence the coding regions of the entire genome, with the aim to identify rare causative and high penetrance mutations in single genes.

EWAS for ICH were performed using 673 patients with ICH and 9158 controls in Japanese individuals. Multivariable logistic regression analysis with adjustment for age, sex and the prevalence of hypertension revealed that rs138533962 [G/A (R379C)] of *STYK1* was significantly associated with ICH, with the minor A allele representing a risk factor for this condition. The relation of *STYK1* to ICH may be attributable to an effect of this gene on the remodeling of blood vessels in the brain, although the molecular mechanism underlying this association remains to be determined [86]. The same group performed EWAS for 261 early-onset ICH patients (≤65 years) vs. 5742 controls and identified *DNTTIP2* (rs3747965 in chromosome 1) and *FAM205A* (rs3739881 in chromosome 9) as susceptibility loci for ICH. Given that the two genes associated with ICH in the present study were not related to intermediate phenotypes, the functional relevance of the association of these genes with ICH remains to be elucidated [87]. A previous study, performed mainly in patients with European ancestry, using exome array, did not identify any rare coding variants for ICH [88].

## 5. Conclusions and Future Directions

Translating previously mentioned discoveries into clinical practice remains a challenge. Further investigations are required to validate those findings. Efforts are now being made to expand genetic studies to non-European ethnic groups, which may have different risk factor profiles. Understanding how genetic risk factors vary across race/ethnicity with large multi-ancestry GWAS, may highlight novel underlying disease mechanisms and identify populations who may be particularly responsive to specific prevention strategies. To address this, projects such as The Ethnic/Racial Variations of Intracerebral Hemorrhage (ERICH) are currently being conducted [89].

Combining genomics information with epigenomics, transcriptomics, proteomics, and metabolomics data offers a unique opportunity to enhance our understanding of the pathological processes related to ICH. The information obtained through these different omic approaches can be integrated—referred to as integromics—becoming an essential step for the design of experimental studies.

The ultimate understanding and therapeutic strategies for ICH must encompass a spectrum of novel technologies based on epigenetic regulatory mechanisms (such as RNA-based therapies) and recent genome-editing tools, with the potential to transform ICH management and significantly improve patient’s outcomes [53,65].

## Figures and Tables

**Figure 1 ijms-23-06479-f001:**
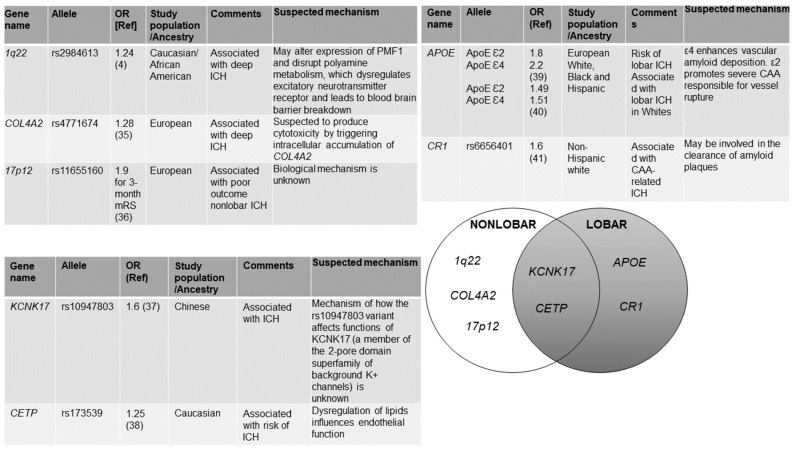
Genetic variants associated with primary intracerebral hemorrhage from genome wide association studies (GWAS). We identified references for this review by searching PubMed from January 2010, to December 2021, using combinations of the terms: “intracerebral hemorrhage” and “GWAS”. We restricted our search to articles published in English. We also found relevant papers by searching authors. We selected material for inclusion on the basis of quality and relevance. In cases of multiple positive reports, only the initial positive study is cited [4,35,36,37,38,39,40,41].

**Table 1 ijms-23-06479-t001:** Summary of microRNAs differentially expressed in patients with primary intracerebral hemorrhage respect to healthy controls, in more than two independent studies. Papers that lacked quantitative analysis were excluded. Based on the systematic review performed by Fullerton and co-workers [67].

ICH Phenotype Studied (n)	MiRNA	Upregulated or Downregulated	Study Population/Ancestry [Ref]
HE in patients with ICH (*n* = 30) vs. Non-HE in ICH patients (*n* = 49) HE-ICH vs. healthy controls (*n* = 30)	miRNA-29c miRNA-29c	Upregulated	China [72]
Hematoma volume (*n* = 33) vs. healthy controls (*n* = 18) Perihematomal edema	miRNA-126 miRNA-146a miRNA-let-7a miRNA-26a miRNA-126	Downregulated	China [73]
ICH patients (*n* = 30) vs. control (*n* = 30) Perihematomal edema	miRNA-23a-3p miRNA-130a miRNA-26a miRNA-146a miRNA- 23a-3p	Upregulated Downregulated Upregulated	China [74]
ICH patients (*n* = 80) vs. healthy control (*n* = 30)	miRNA-155	Upregulated	China [75]
ICH patients (*n* = 106) vs. healthy control (*n* = 50)	miRNA-145 miRNA-223, and miRNA-155 miRNA-181b	Upregulated Downregulated	China [76]

Abbreviations: ICH = intracerebral hemorrhage; MiRNA = micro RNA; HE = hematoma enlargement; vs. = versus.

## Data Availability

Not applicable.

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
