# Peer review of "Genetics and Epigenetics of Spontaneous Intracerebral Hemorrhage"

_ijms, 2022, doi:10.3390/ijms23126479_

Round 1

Reviewer 1 Report

Giralt-Steinhauer et al. w summarized how current knowledge on genetics and epigenetics of this devastating stroke subtype were contributing to improve the understanding of ICH pathophysiology, and their potential role in developing therapeutic strategies. It is interesting study, well-written. It focuses on the very importgant topic. I have some minor suggestion for authors. It would be nice to create the Figure which summarize the patomechanism of genes implicationg on ICH. It would be nice to refer to the genetic variants associated with ischemic stroke (Milanowski et al. Pharmacogenomics. 2016 Jun;17(8):953-71.). It wolud be also good to add some general information about subtypes of ICH (Neurol Neurochir Pol 2021;55(5):450-461). Therefore, I recommend minor revision.

Reviewer 2 Report

Proposed paper is interesting and well written. Authors done a complete litterature review with an exellent final result. However, I have one comments that can improve the paper furthermore.

One of the determinant of ICH is also blood pressure (cfr and consider to cite: High Blood Press Cardiovasc Prev. 2018 Jun;25(2):177-189.) that is totally skipped. Are specific study on ICH genomic and epigenomic and blood pressure relationship present in litterature? if yes please discuss also this point. If no, please add a brief paragraph on the importance of blood pressure in ICH pathogenesis in the introdution.
